# A Survey on Deep Graph Generation: Methods and Applications

**Yanqiao Zhu***
UCLA
yzhu@cs.ucla.edu

**Yuanqi Du***
Cornell
yd392@cornell.edu

**Yinkai Wang***
Tufts
yinkai.wang@tufts.edu

**Yichen Xu**
EPFL
yichen.xu@epfl.ch

**Jieyu Zhang**
UW
jieyuz2@cs.washington.edu

**Qiang Liu**
CASIA
qiang.liu@nlpr.ia.ac.cn

**Shu Wu**
CASIA
shu.wu@nlpr.ia.ac.cn

## Abstract

Graphs are ubiquitous in encoding relational information of real-world objects in many domains. Graph generation, whose purpose is to generate new graphs from a distribution similar to the observed graphs, has received increasing attention thanks to the recent advances of deep learning models. In this paper, we conduct a comprehensive review on the existing literature of deep graph generation from a variety of emerging methods to its wide application areas. Specifically, we first formulate the problem of deep graph generation and discuss its difference with several related graph learning tasks. Secondly, we divide the state-of-the-art methods into three categories based on model architectures and summarize their generation strategies. Thirdly, we introduce three key application areas of deep graph generation. Lastly, we highlight challenges and opportunities in the future study of deep graph generation. We hope that our survey will be useful for researchers and practitioners who are interested in this exciting and rapidly-developing field.

## 1 Introduction

Graphs are ubiquitous in modeling relational and structural information of real-world objects in many domains, ranging from social networks to chemical compounds. Generating realistic graphs therefore has become a key technique to advance a variety of fields [1]. For example, in drug discovery and chemical science, a fundamental yet challenging task is to generate novel, realistic molecular graphs with desired properties (e.g., high drug-likeness and synthesis accessibility). Due to the discrete and high-dimensional nature of graph structures, exploring the drug-like molecules on the chemical space involves combinatorial optimization, as the size of the space is estimated to be $10^{60}$ [2]. In this application, graph generation algorithms could help expedite the drug discovery process by discovering new candidate molecules with desired properties.

Traditional graph generation models assume that real graphs obey certain statistical rules. These models compute hand-crafted statistical features of existing graphs and generate new graphs with similar features. However, this assumption oversimplifies the underlying distributions of graphs and is thus not capable of capturing complex graph distributions in real scenarios. For example, the Barabási-Albert model [3] assumes similar graphs follow the same empirical degree distribution, but this model fails to capture other aspects (e.g., community structures) of real-world graphs. Recently, there is an increasing interest in developing deep models for graph-structured data which enables effective complex graph generation. To name a few, GraphRNN [4] treats graph generation as a sequential generation problem and generates nodes and edges step by step; GraphVAE [5] proposes a

---

*Equal contribution; listing order determined by coin flipping.

Y. Zhu et al., A Survey on Deep Graph Generation: Methods and Applications. *Proceedings of the First Learning on Graphs Conference (LoG 2022)*, PMLR 198, Virtual Event, December 9–12, 2022.

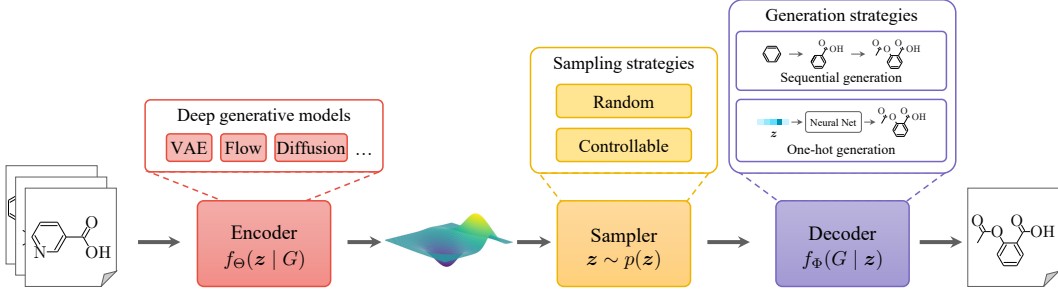

**Figure 1:** An overview of deep graph generation approaches: the encoder maps observed graphs into a stochastic distribution; the sampler draws latent representations from that distribution; the decoder receives latent codes and produces graphs.

VAE-based graph generative model and generates new graphs in a one-shot manner; MoFlow [6] designs an invertible mapping between the input graph and the latent space and generate the graph (node feature and edge feature matrices) in one single step; MolGAN [7] designs a GAN-based graph generative model where a discriminator is used to ensure the properties of the generated graphs; GDSS [8] designs a score-based graph generative model which adds Gaussian noise to both node features and structures and reconstructs from Gaussian noise to obtain generated graphs during inference. Additionally, many other graph generative methods are utilized for deep graph generation [9, 10, 11, 12, 13, 14].

Up to now, although many recent survey works have reviewed deep graph learning approaches, most of them focus on graph representation learning [15, 16, 17] and little attention has been paid to systematically review graph generation techniques. Two other surveys [18, 19] mostly focus on the generation process and generative models while we focus on the entire spectrum of graph generation from generative models, sampling strategies, to generation strategies. We also discuss state-of-the-art techniques such as diffusion and score-based generative approaches. By categorizing and discussing existing models of graph generation, we envision that this work will elucidate core design considerations, discuss common approaches and their applications, and identify future research directions in graph generation.

The remaining of this paper is structured as follows. Firstly, we formulate the problem of graph generation and differentiate it from several closely related graph learning tasks (Section 2). Then, we give an algorithm taxonomy that groups existing methods into three categories: latent variable approaches, reinforcement learning approaches, and other graph generation models (Section 3). In this section, we present a general framework, discuss common generation strategies in detail, and introduce representative work of each type. Thirdly, we demonstrate how graph generation could lead to great success in three promising application areas (Section 4). Finally, we conclude the paper with challenges and future promises of deep graph generation (Section 5).

## 2 Problem Definition

We define a graph by a quadruplet $G = (\mathcal{V}, \mathcal{E}, \boldsymbol{X}, \mathbf{E})$, where $\mathcal{V}$ is the vertex set, $\mathcal{E} \subseteq \mathcal{V} \times \mathcal{V}$ is the edge set, $\boldsymbol{X} \in \mathbb{R}^{N \times D}$ is the node feature matrix, $\mathbf{E} \in \mathbb{R}^{N \times N \times F}$ is the edge attributes, and $D, F$ are the feature dimensionality. Given a set of $M$ observed graphs $\mathcal{G} = \{G_i\}_{i=1}^{M}$, graph generation learns the distribution of these graphs $p(\mathcal{G})$, from which new graphs can be sampled $G_{\text{new}} \sim p(\mathcal{G})$.

**Related problems.** In the regime of graph learning, there are three problems that are closely related to, but different from, deep graph generation. Here, we succinctly compare them with graph generation and we refer readers of interest to relevant surveys for a comprehensive understanding of these areas.

- **Link prediction** [20, 21] aims to predict the possibility of the missing links between a pair of nodes in a graph. Some generative link prediction models estimate the distribution of edge connectivity, and thus could be used for graph generation as well.

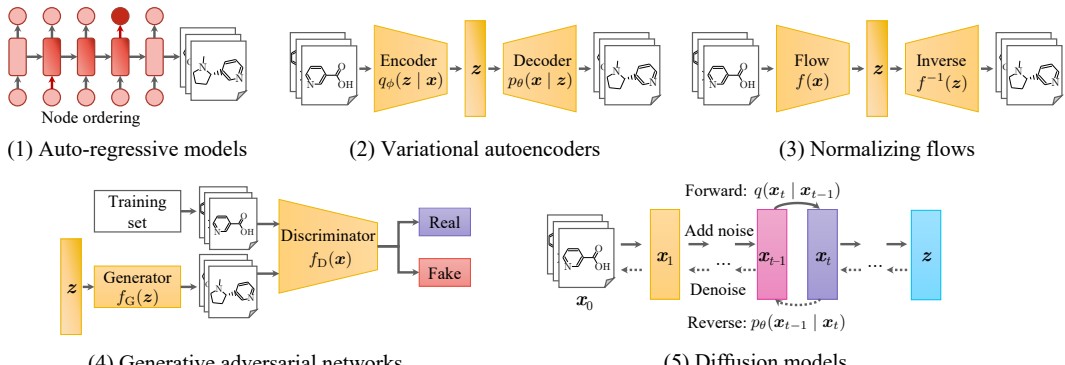

**Figure 2:** A summary of graph generative models for deep graph generation, including (1) auto-regressive models, (2) variational autoencoders, (3) normalizing flows, (4) generative adversarial networks, and (5) diffusion models.

- **Graph structure learning** [22, 23] simultaneously learns an optimized graph structure along with representations for downstream tasks. Unlike graph generation that aims to generate new graphs, the purpose of graph structure learning is to improve the given noisy or incomplete graphs.

- **Generative sampling** [24, 25, 26] learns to generate subsets of nodes and edges from a large graph. As most graph generative models do not scale to large single-graph datasets such as citation networks, graph generative sampling could serve as an alternative approach to generate large-scale graphs by sampling subgraphs from a large graph and reconstructing a new graph.

- **Set generation** [27, 28] seeks to generate set objects, such as point clouds or 3D molecules, which is similar to graph generation in that graphs are also set objects. In this survey, we only focus on graph generation whose objective concerns with generation of both the nodes and edges matrices, whereas set generation typically does not consider edge features. Nevertheless, we recognize that several set generation methods share significant similarities with graph generation.

## 3 Algorithm Taxonomy

For deep graph generation, we present an encoder–sampler–decoder pipeline, as shown in Figure 1, to characterize most existing graph generative models in a unified framework. Here, the observed graphs are first mapped into a stochastic low-dimensional latent space, with latent representations following a stochastic distribution. A random sample is drawn from that distribution and then passed through a decoder to restore graph structures, which are typically represented in an adjacency matrix as well as feature matrices. Under this framework, we organize various methods around three key components:

**The encoder.** The encoding function $f_\Theta(z \mid G)$ represent discrete graph objects as dense, continuous vectors. To ensure the learned latent space is meaningful for generation, we employ probabilistic generative models (e.g., variational graph neural networks) as the encoder. Formally, the encoder function $f_\Theta$ outputs the parameters of a stochastic distribution following a prior distribution $p(z)$.

**The sampler.** Consequently, the graph generation model samples latent representations from the learned distribution $z \sim p(z)$. In graph generation, there are two sampling strategies: random sampling and controllable sampling. **Random sampling** refers to randomly sampling latent codes from the learned distribution. It is also called distribution learning in some literature [29]. In contrast, **controllable sampling** aims to sample the latent code in an ultimate attempt to generate new graphs with desired properties. In practice, controllable sampling usually depends on different types of deep generative models and requires an additional optimization term beyond random generation.

**The decoder.** After receiving the latent representations sampled from the learned distribution, the decoder restores them to graph structures. Compared to the encoder, the decoder involved in the

graph generating process is more complicated due to the discrete, non-Euclidean nature of graph objects. Specifically, the decoders could be categorized into two categories: sequential generation and one-hot generation. **Sequential generation** refers to generating graphs in a set of consecutive steps, usually done nodes by nodes and edges by edges. **One-shot generation**, instead, refers to generating the node feature and edge feature matrices in one single step.

It should be noted that not all methods include all of the components discussed in this framework. For example, Generative Adversarial Networks (GANs) often do not include a specific encoder component.

## 3.1 Deep Generative Models

At first, we discuss the following five representative deep generative models, which aims to learn the probability distribution of graphs so that we can sample new graphs from it.

**Auto-Regressive models (AR).** AR models factorize a joint distribution over $N$ random variables via the chain rule of probability. Specifically, this model factorizes the generation process as a sequential step which determines the next step action given the current subgraph. The general formulation of AR models is as follows:

$$p(G^\pi) = \prod_{i=1}^{N} p(G_i^\pi \mid G_1^\pi, G_2^\pi, \cdots, G_{i-1}^\pi) = \prod_{i=1}^{N} p(G_i^\pi \mid G_{<i}^\pi), \tag{1}$$

where $G_{<i}^\pi = \{G_1^\pi, G_2^\pi, \cdots, G_{i-1}^\pi\}$ is the set of random variables in the previous steps. Since AR works like sequential generation, applying AR models requires a pre-specified ordering $\pi$ of nodes in the graph.

**Variational Autoencoders (VAEs).** The VAE [30] estimates the distributions of graphs $p(\mathcal{G})$ by maximizing the Evidence Lower BOund (ELBO) as follows:

$$\mathcal{L}_{\text{VAE}} = \mathbb{E}_{z \sim q_\phi(z|G)} \log(p_\theta(G \mid z)) - D_{\text{KL}}(q_\phi(z \mid G) \parallel p_\theta(z))), \tag{2}$$

where the former term is known as the reconstruction loss between the input and the reconstructed graph, while the latter is the disentanglement enhancement term that drives $q_\phi(z \mid G)$ to the prior distribution $p_\theta(z)$, usually a Gaussian distribution. The encoder $p(z \mid G)$ and decoder $q(G \mid z)$ are typically parametrized by graph neural networks (e.g., GCN [31], GAT [32]).

**Normalizing Flows.** Normalizing flow estimates the density of graphs $p(\mathcal{G})$ directly with an invertible and deterministic mapping between the latent variables and the graphs via the change of variable theorem [33, 34]. A typical instance of flow-based models takes the following form:

$$p(G) = p(z) \left| \det\left(\frac{\partial f^{-1}(G)}{\partial G}\right) \right|, \tag{3}$$

where $\frac{\partial f^{-1}(\cdot)}{\partial \cdot}$ is the Jacobian matrix. As the encoder $f(G)$ needs to be invertible, the decoder is essentially $f^{-1}(z)$. Then, normalizing-flow-based models are usually trained by minimizing the negative log-likelihood over the training data $\mathcal{G}$.

**Generative Adversarial Networks (GANs).** The GAN model is another type of generative models, especially popular in the computer vision domain [35]. It is an implicit generative model, which *learns* to sample real graphs. GAN consists of two main components, namely, a generator $f_{\text{G}}$ for generating realistic graphs and a discriminator $f_{\text{D}}$ for distinguishing between synthetic and real graphs. Formally, its training objective is a min-max game as follows:

$$\min_{f_{\text{G}}} \max_{f_{\text{D}}} \mathcal{L}_{\text{GAN}}(f_{\text{G}}, f_{\text{D}}) = \mathbb{E}_{G \sim p(G)}[\log f_{\text{D}}(G)] + \mathbb{E}_{z \sim p(z)}[\log(1 - f_{\text{D}}(f_{\text{G}}(z)))]. \tag{4}$$

**Diffusion models.** Diffusion or score-based generative models are a new class of generative models inspired by nonequilibrium thermodynamics [36, 37, 38]. Diffusion models contain two processes, the forward and the reverse diffusion process. The forward diffusion process constantly adds noise to the data sample $x_0$, while the reverse diffusion process recreates the true data sample from a

Gaussian noise input $\boldsymbol{x}_T \sim \mathcal{N}(\boldsymbol{0}, \boldsymbol{I})$. Specifically, the forward diffusion process from step $(t-1)$ to $t$ is defined as:

$$q(\boldsymbol{x}_t \mid \boldsymbol{x}_{t-1}) = \mathcal{N}(\boldsymbol{x}_t;\ \sqrt{1-\beta_t}\boldsymbol{x}_{t-1},\ \beta_t\boldsymbol{I}), \tag{5}$$

$$q(\boldsymbol{x}_{1:T} \mid \boldsymbol{x}_0) = \prod_{t=1}^{T} q(\boldsymbol{x}_t \mid \boldsymbol{x}_{t-1}), \tag{6}$$

where $\beta_t \in (0,1)$ controls the step size. Note that the reverse diffusion process $q(\boldsymbol{x}_{t-1} \mid \boldsymbol{x}_t)$ will also be Gaussian if $\beta_t$ is small enough. However, since $q(\boldsymbol{x}_{t-1} \mid \boldsymbol{x}_t)$ is intractable, we learn a model $p_\theta$ to approximate these conditional probabilities, which is defined as:

$$p_\theta(\boldsymbol{x}_{t-1} \mid \boldsymbol{x}_t) = \mathcal{N}(\boldsymbol{x}_{t-1};\ \boldsymbol{\mu}_\theta(\boldsymbol{x}_t, t),\ \boldsymbol{\Sigma}_\theta(\boldsymbol{x}_t, t)), \tag{7}$$

We use the variational lower bound to optimize the negative log-likelihood:

$$-\log p_\theta(\boldsymbol{x}_0) \leq \mathbb{E}_{q(\boldsymbol{x}_{1:T}|\boldsymbol{x}_0)}\left[\log \frac{q(\boldsymbol{x}_{1:T} \mid \boldsymbol{x}_0)}{p_\theta(\boldsymbol{x}_{0:T})}\right], \tag{8}$$

where

$$p_\theta(\boldsymbol{x}_{0:T}) = p(\boldsymbol{x}_T)\prod_{t=1}^{T} p_\theta(\boldsymbol{x}_{t-1} \mid \boldsymbol{x}_t). \tag{9}$$

The final objective takes expectation over $q(\boldsymbol{x}_0)$ on both sides of Equation (8):

$$\mathcal{L}_{\text{VLB}} = \mathbb{E}_{q(\boldsymbol{x}_{0:T})}\left[\log \frac{q(\boldsymbol{x}_{1:T} \mid \boldsymbol{x}_0)}{p_\theta(\boldsymbol{x}_{0:T})}\right] \geq -\mathbb{E}_{q(\boldsymbol{x}_0)}\log p_\theta(\boldsymbol{x}_0). \tag{10}$$

## 3.2 Sampling Strategies

After learning a latent space for representing the input graphs, we sample new latent code so as to manipulate the graphs to be generated. The sampling strategies could be divided into two categories, random sampling and controllable sampling. **Random sampling** simply draws latent samples from the prior distribution, in which the model learns to approximate the distribution of the observed graphs. The latter, on the contrary, samples new graphs with controls (i.e. desired properties). Therefore, for latent variable approaches, random sampling is relatively trivial, while controllable sampling usually requires extra effort in algorithm design.

Controllable generation usually manipulates the randomly sampled $\boldsymbol{z} \sim p(\boldsymbol{z})$ or the encoded vector $\boldsymbol{z} \sim p(\boldsymbol{z} \mid G)$ in the latent space to obtain a final representation vector $\widetilde{\boldsymbol{z}}$, which is later decoded to a graph with expected properties. There are three types of commonly used approaches:

- **Disentangled sampling** factorizes the latent vector $\boldsymbol{z}$ with each dimension $\boldsymbol{z}_n$ focusing on one property $p_n$, following the disentanglement regularization that encourages the learnt latent variables to be disentangled from each other. Therefore, varying one latent dimension $\boldsymbol{z}_n$ of the latent vector $\boldsymbol{z}$ will lead to property change in the generated graphs.

- **Conditional sampling** introduces a conditional code $\boldsymbol{c}$ that explicitly controls the property of generated graphs. In this case, the final representation $\widetilde{\boldsymbol{z}}$ is usually the concatenation of $\boldsymbol{z}$ and $\boldsymbol{c}$.

- **Traverse-based sampling** searches over the latent space by directly optimizing the continuous latent vector $\boldsymbol{z}$ to obtain $\widetilde{\boldsymbol{z}}$ with specific properties or uses heuristic-based approach (e.g., linear or nonlinear interpolation from $\boldsymbol{z}$ to obtain $\widetilde{\boldsymbol{z}}$), to control the property of the generated graphs.

## 3.3 Generation Strategies

Finally, the decoder restores the latent code back to graph structures. Due to the discrete, high-dimensional, and unordered nature of graph data, the resulting non-differentiability hinders the backpropagation of the graph decoder, unlike continuous generation in image or audio domains. To address this issue, existing works take two types of generation strategies for graph generation, one-shot generation and sequential generation.

**One-shot generation.** One-shot generation usually generates a new graph represented in an adjacency matrix with optional node and edge features in one single step. It is achieved by feeding the latent representations to neural networks to obtain the adjacency and feature matrices. In practice, various neural networks could be utilized, including Convolutional Neural Networks (CNN), Graph Neural Networks (GNN), Multi-layer Perceptron (MLP) [39, 40, 41], according to different types of feature matrices to be generated. For example, Du et al. [42] utilize 1D-CNN to decode the node feature and 2D-CNN for the edge feature, and Flam-Shepherd et al. [40] jointly utilize a GNN and a MLP for the decoder. The advantage of one-shot generation is that it generates the whole graph in a single step without sequential dependency on node ordering, while it has to set a predefined maximum number of nodes and may suffer from low scalability as it scales as $O(N^2)$ with respect to $N$ nodes in the graphs.

**Sequential generation.** In contrast to one-shot generation, sequential generation generates a graph consecutively in a few steps. As there is no ordering naturally defined for graphs, sequential generation has to follow a certain ordering of nodes for the generation. This is usually done by generating a probabilistic node feature and edge feature matrices while sampling step by step from the matrices following a predefined node ordering (e.g., breadth-first search [43, 44]). Sequential generation enjoys the benefit of flexibility, especially when the number of nodes to generate is unknown beforehand. Therefore, it could be easily combined with constraint checking in each of the generation steps, when the graph to be generated should obey certain restrictions. However, when generating a large graph with a long sequence, the error will accumulate at each step, possibly resulting in discrepancies in the final generated and observed graphs.

## 3.4 Discussion: Permutation Invariance and Equivariance

Graphs are inherently invariant with respect to permutation, which means that any arbitrary permutation on nodes should result in the same graph representation. As such, graph generative models need to model permutation-invariant graph distributions. Under certain mild conditions, it is possible for different generation models (e.g., GANs/VAEs [45], normalizing flows [46], diffusion or score-based generative models [47], and energy-based models [48]) to achieve this goal. In most of these cases, a simple graph neural network with a *permutation-equivariant encoder*, e.g., GCN [31] and GAT [32], will suffice. This permutation-equivariance property ensures that when given a permuted graph, it produces equivalently permuted node representation vectors. However, auto-regressive models often require a node ordering, e.g., breath-first search in GraphRNN [4]. It is thus nontrivial for them to achieve permutation invariance.

## 3.5 Representative Work

In this subsection, we succinctly discuss a few representative works in each type of generative models with an emphasis on how they handle controllable generation.

**Auto-regressive models.** AR models naturally generate graphs in a sequential way, while it requires a specified node ordering. GraphRNN [4] leverages breath-first search to determine the node ordering and generates nodes and its associated edges sequentially. In contrast, Bacciu et al. [49], Goyal et al. [50], Bacciu and Podda [51] design edge-based auto-regressive models that generate each edge and the nodes it connects sequentially. Additionally, since the auto-regressive model can determine the action for the next step given the current subgraph, by formulating graph generation as a sequence of the decision-making processes, it is commonly used as a policy network together with Reinforcement Learning (RL). MolecularRNN [52] designs an RL environment with an auto-regressive model as the policy network to generate new nodes and edges sequentially for new graphs. Rewards are designed for controllable generation that generates graphs with desired properties.

**Variational autoencoders.** VAE is a simple yet flexible framework and could be adopted for controllable sampling by modifying the loss function to enforce latent variables to be correlated with properties of interest [53, 54, 55, 56]. MDVAE [54] designs a monotonic constraint between the latent variables and the properties such that increasing the values of latent variables leads to increasing the values of the properties. PCVAE [56] learns an invertible mapping between the latent variables and the properties in which generating graphs with desired properties is as trivial as inverting the mapping function. This approach could also proceed in an unsupervised fashion and has demonstrated controllability over graph properties [41, 42, 57, 58].

The other approaches [59, 60, 61] leverage the learned continuous and meaningful latent space with Bayesian optimization, search for latent vectors optimizing specific properties, and then decode the graphs from the latent vectors. Du et al. [53] also introduces a new method that aims to control the properties of the generated molecules via a smooth linear interpolation over the latent space.

Additionally, optimization-based methods are also developed to search latent vectors that possess desired or optimal molecular properties. JT-VAE [61] performs Bayesian optimization on the latent vector searching for molecular graphs with optimal properties. Kajino [62] circumvents the designs for a complex network to generate valid graphs by introducing a graph grammar that encodes the hard chemical constraint for molecular graphs. Yang et al. [63] combine a conditional VAE-based model with adversarial training to incorporate semantic contexts in graph generation. Zhang et al. [64] work on the generation of directed acyclic graphs with an asynchronous message passing scheme. Samanta et al. [65] incorporate the 3D coordinates of molecular graphs into the model and thus is capable of generating both 2D graphs and 3D coordinates of the molecules. Lim et al. [66] especially take care of one application scenario where a predefined subgraph is given and the rest of the graph needs to be completed. Li et al. [67] introduce a new perspective to view the reconstruction of the VAE-based model in graph generation as a balanced graph cut.

**Normalizing flows.** Normalizing flow is also a commonly used model in deep graph generation. GraphNVP [68] first adapts normalizing flow to graph generation which encodes graph node feature and edge feature matrices in the latent space and then reverses the flow to generate the graphs represented by the node feature and edge features matrices. Nevertheless, it adopts one-shot generation on molecular graph generation while failing to generate fully-valid molecules in the absence of the validity constraint. MoFlow [6] also adopts one-shot generation but further designs a valency correction as a post-processing step that corrects the generated invalid molecular graphs. This line of work demonstrates the advantage of sequential generation in the sense that they are able to generate syntactically valid new graphs, while one-shot generation may require post-processing since it does not impose any constraint on the generated graphs. For controllable sampling, flow-based methods also learn a continuous latent space and adopt optimization-based methods on the latent space to search for latent vectors with expected properties. MoFlow [6] adopts the regression optimization to optimize the latent vectors for desired properties. GraphDF [69] challenges the commonly used approach that learns a continuous latent space for graph generation and designs a normalizing flow-based approach that learns discrete latent variables.

**Generative adversarial networks.** GAN-based models by design allow easy implementation of controllable sampling, e.g., by introducing a property discriminator for desired properties. MolGAN [7] learns to sample the probability matrix for the node feature and edge feature, respectively. It directly generates new graphs by taking the maximum likelihood of the nodes and edges. It also designs a reward discriminator which determines the property score of the generated graphs. However, there remains a paucity of GAN-based graph generative models most likely due to the difficulty of designing generators. Guarino et al. [70] design a GAN-based model that learns hierarchical representations of graphs in the discriminator network. Jin et al. [71] leverage adversarial training that discriminates the generated graphs and the expected graphs. Pølsterl and Wachinger [72], Maziarka et al. [73] introduce a cycle-consistency loss in the GAN-based model for graph generation. Fan and Huang [74] propose a conditional GAN model for graph generation. Gamage et al. [75] propose a GAN-based model that focuses on learning higher-order structures or motifs for the graph generation. Yang et al. [76] generate target relation graphs modeling the underlying interrelationships among time series.

**Diffusion models.** Diffusion or score-based generative models allow the generation of high-quality data involving various data modalities, including images [77], audios [78], point clouds [27], etc. Recently, diffusion models have been adopted to graph-structured data generation as well [8, 47]. Specifically, Niu et al. [47] design a score-based generative model that estimates the score of the graph topology (adjacency matrix) and samples new graph topology by leveraging Langevin dynamics. Its follow-up work GDSS [8] and DiGress [79] further consider generating the node feature vectors and graph topology together. Unlike other diffusion models that train the energy function using a score-matching objective function, GraphEBM [48] resorts to contrastive divergence and generates new graphs by leveraging Langevin dynamics [80].

## 3.6 Other Approaches

While many models involve only one type of generative models for graph generation, it is also possible to develop methods with hybrid generative models, enjoying the advantages of ensemble models. For example, GraphAF [81] adopts normalizing flow in an auto-regressive model framework. Additionally, other optimization or searching methods which directly sample from the data space rather than the latent space are also introduced for graph generation [47, 48, 82, 83, 84, 85, 86, 87]. MARS [13] employs Markov Chain Monte Carlo sampling (MCMC) that iteratively edits the graphs to optimize the objective (i.e. desired property). DST [9] directly optimizes the graph representation (i.e. node feature matrix and edge feature matrix) while optimizing the property of the newly generated graph.

# 4 Applications

In this section, we discuss the real applications of graph generation. Specifically, we focus on three concrete examples, molecule design, protein design, and program synthesis. We illustrate their formulations in graph generation and how graph generation techniques could lead to success in various real-world applications.

## 4.1 Molecule Design

In molecule generation, there are two goals for generative design: (1) graph generative methods should generate syntactically valid molecules and (2) the generated molecule should possess certain properties. Sequential generation strategy could ensure the validity of the generated molecules by incorporating a valency check in each intermediate generation step. GraphAF [81] designs an auto-regressive flow that takes an iterative sampling process and allows for valency check in each step, thus achieving high validity in the generated molecules. However, one-shot generation may suffer from the low validity of the generated molecules. GraphNVP [68] and GRF [88] first introduce normalizing-flow-based models into molecular graph generation and design invertible mapping layers for node features and edge features, while both suffering from the low validity of the generated molecules due to the lack of valency constraint within one-hot generation. Moflow [6] improves over GraphNVP with a post-valency correction step which solves the low-validity issues of the generated molecules. For controllable generation, VAE- and Flow-based methods [6, 61] usually connect with traverse-based sampling that searches over the learned continuous latent space for vectors/molecules with desired properties. For GAN- and VAE-based methods [7, 54], they are suitable for conditional generation (i.e. conditional sampling), where the latent code could control the properties of the generated molecules. Furthermore, reinforcement learning approaches can achieve controllable generation for molecule design. They are typically used in conjunction with another generative model (e.g., AR, GANs) to generate molecules with desired properties by designing appropriate reward functions [7, 14, 81].

## 4.2 Protein Design

Protein design is another critical application of graph generation. Protein is naturally a sequence of amino acids and could be represented as graphs by constructing a pairwise contact map based on 3D structure data, because the 3D structures of protein determine its functions. Specifically, the contact map establishes edge connectivity when two nodes (residues) have contacted with each other. In protein generation, early work [89] represents the pairwise contact map as grid data and processes it with CNNs. However, representing the protein contact as a grid only considers adjacent residues as neighbors, while graph representations could capture more local contact information [57]. Representative work [90] designs an auto-regressive model for protein sequence design given the 3D structures represented by graphs. Recently, Guo et al. [57] design a VAE-based graph generative model that generates new protein contact maps and then decodes the 3D structure. Jin et al. [91] introduce an iterative refinement GNN model that designs both the sequences and structures of the Complementarity-Determining Regions (CDRs) of antibodies.

## 4.3 Program Synthesis

Graph generation can also be applied in program synthesis. Program synthesis aims at generating programs from specifications consisting of natural language description and input output samples.

Traditional methods formulate program synthesis as a sequence-to-sequence problem and employ language modeling techniques from the NLP community [92, 93]. However, unlike natural language data, programming languages are well structured by their nature. To model the intrinsic structures underlying programs, researchers propose the notion of program graphs [94, 95] that incorporate the knowledge from program syntax and semantics. Specifically, the program graph can be constructed from the Abstract Syntax Trees (AST) of programs with additional edges based on program semantics. Regarding graph generation for programs, a natural idea is to synthesize programs by generating ASTs [83, 96, 97, 98]. To enforce the validity of generated programs, most existing methods take the sequential generation strategy: the model will choose one grammar rule to expand one non-terminal node in the partially-generated graph. To determine the order of generation, most approaches seek to expand the left-most, bottom most non-terminal node [83, 98, 99]. Take several representative works as examples; Brockschmidt et al. [83] propose to augment the partially-generated ASTs with additional syntactic and semantic connections to incorporate prior knowledge from static program analysis into the generation process. Brockschmidt et al. [83], Dai et al. [100] go beyond context-free grammars and employ attributed grammars as the generation framework in order to encourage the semantic validity of resulting program graphs.

## 5   Challenges and Opportunities

Despite the impressive progress that has been made in the field of deep graph generation, there is still ample room for further development of these methods and their applications. In this section, we highlight some of the challenges and limitations of prior work in this area, and discuss potential directions for future research.

**Evaluation pipeline.**   The evaluation of graph generative models is one of the main bottlenecks that hinder the advances of the field of ever-increasing complexity [101, 102]. Like generation in other domains, graph generation is hard to evaluate due to the absence of ground-truth labels. Therefore, current evaluations mostly depend on prior knowledge (i.e. graph statistics, properties) about the graphs, while the real-world applications typically require expensive evaluations, e.g., wet-lab experiments for molecule design. Furthermore, the selected statistics and properties are typically task-specific, i.e., some statistics are important for one type of graph while may be irrelevant for another type of graph. Further work is required to establish proper evaluation metrics and pipeline for graph generation models.

**Graph properties/rules design.**   Currently, the graph properties or rules utilized for controllable generation are quite simple and limited to a small set. For example, in molecular graph design, the molecular properties utilized are usually simple molecular descriptors, while expensive and real-world drug discovery oracles, e.g., synthesis accessibility, protein-binding affinity score, could be studied in the future.

**Diverse graph types.**   Graph is ubiquitous in the world and many data could be interpreted as graph structures, e.g., spatial networks (such as molecules, social networks, and circuit networks), temporal graphs (such as traffic networks and dynamical system simulations), etc. Yet, different types of graphs are usually largely distinct. However, the current study on graph generation mostly focuses on molecular graphs while ignoring the diversity of real graph data, partially due to the low availability of large repositories of graph data in many domains.

**Scalability.**   The scalability of graph generative methods is usually bounded to the complexity of the encoder and decoder design. A few effort have been made in this direction: Dai et al. [103], Kawai et al. [104] leverage the sparsity of graph structures and parallel training for auto-regressive models. However, the current decoder design with an one-hot generation strategy still has poor scalability due to its $O(N^2)$ complexity with regard to $N$ nodes. In light of the fact that many real-world graph data, e.g., proteins, materials, etc., are large in scales, designing scalable encoders and decoders is critical yet under-explored.

**Interpretability.**   Even though graph generation is capable of generating new graphs, the generation process has low interpretability [105]. To improve the transparency of the generation model, we could consider the following aspects of interpretability: interpreting a series of decision-making process to improve certain property of a graph, interpreting how graph generative models learn the latent space

that control the properties of the generated graphs, and interpreting the complex properties of the graph data with respect to the graph structures.

## 6 Concluding Remarks

In this paper, we present a comprehensive review of deep graph generation models and applications. Specifically, we formulate the deep graph generation task through a unified encoder–sampler–decoder framework and present an algorithm taxonomy with three key components. Following that, for each component, we discuss the common graph generation techniques and their key characteristics in detail. Thereafter, we focus on three application areas in which deep graph generation plays an important role. Finally, we highlight challenges in current studies and discuss future research directions of deep graph generation.

## Supplementary Material

For readers of interest, in Appendix A, we summarize representative deep graph generation models on their encoding architectures, generation strategies, sampling strategies, graph types involved, and application areas; in Appendix B, we provide an overview of the evaluation metrics and commonly-used datasets to assess the performance of various deep graph generation methods in the literature.

## Acknowledgements

This work is supported by National Natural Science Foundation of China (U19B2038).

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

## A    Summary of Advances in Deep Graph Generation

Deep graph generation is a rapidly growing field with many technology advancements and application areas. In this section, we provide a summary of representative works in deep graph generation, organized by methodology and application areas. Table 1 provides a useful overview of the state of the art in deep graph generation, demonstrating how different works in this field use different generative models, generation strategies, and sampling strategies to generate graphs in different domains, such as social networks, biological networks, and chemical networks.

## B    Evaluation Metrics

In this section, we review the evaluation metrics that are used for assessing the performance of the various deep graph generation methods and algorithms discussed in the existing literature. These metrics provide a basis for evaluating the performance of the methods and algorithms on different datasets and allow us to compare and contrast the strengths and limitations of each approach.

To evaluate the quality of the generated graphs, there are two typical sets of metrics distinguished by the goal of generation. For distribution learning or random generation, **statistics-based** metrics compare the distribution of proprieties between the training graphs and the generated graphs and **self-quality-based** metrics evaluate whether the generated graphs are of good quality, such as their validity in the case of molecules [106]. For controllable generation, additional metrics may be used to measure the distance between the generated graphs and the expected graphs, such as the mean squared error between the expected property value and the actual property value. In Table 2, we provide a comprehensive categorization of the various evaluation metrics used in different domains.

**Dataset availability.**    It is important to have access to appropriate benchmarking datasets in order to advance the field of deep graph generation and develop effective methods and algorithms. We additionally summarize commonly-used datasets for graph generation in Table 2. Note that most of the datasets are available at various data repositories, such as GraphGT [106], TDC [107], and PDB [108].

### B.1    Self-Quality-Based Metrics

Common self-quality-based metrics include **validity**, which measures the percentage of valid graphs in the generated set; **novelty**, which measures the percentage of novel graphs not in the training set; and **uniqueness**, which measures the percentage of non-repeated graphs in the generated set.

### B.2    Statistics-Based Metrics

Statistics-based metrics can be further divided into metrics that apply to general graph properties (such as degree distribution, cluster coefficient distribution, and orbit count statistics) and metrics that apply to domain-specific properties.

- **Molecules:** octanol-water partition coefficient (LogP), Quantitative Estimate of Druglikeness (QED), Synthesis Accessibility (SA), and activities against protein targets (e.g., DRD2, JNK3, GSK3$\beta$) [109].

- **Protein:** short-range and long-range contact that measures the distance over pairs of residues, protein perplexity that measures sequence similarity, and fitness that measures mutation effects [57].

- **Traffic networks:** the centrality in a graph based on shortest paths (betweenness centrality), the importance or influence of a node in a network, depending on how many neighbors a node can connect to using just its own connections (broadcast centrality), the centrality base on the intermittent increases and decreases in activity or frequency of traffic (burstiness centrality), the inverse of the lengths of all the shortest paths connecting the node to every other node in the graph (closeness centrality), temporal correlation of nodes (node temporal correlation), and the overall average likelihood that an edge will remain after two successive time steps (temporal correlation) [110].

- **Physical simulation:** prediction error of the future states of physical systems [111].

- **Social networks:** the number of claws on the graph (claw count), the number of wedges on the graph (wedges count), the size of the largest connected component of the graph (LCC), the exponent of the power-law distribution of the graph (PLE), and the number of connected components (N-Component) [112].

- **Skeleton graphs:** Frechet Inception Distance (FID), which calculates the difference between the feature space statistics of actual and synthetic data, and Inception Score (IS), which feeds produced samples to a classification model, which then assesses the output probabilities across all classes [113].

- **Power grids:** the total amount of steps required to travel the shortest distance between any two network nodes (path length), the shortest distance across the network's two furthest nodes (network diameter), the proportion of actual edges to the maximum number of edges that a graph can support (density), the network's ability to be divided into independent, interchangeable components (modularity), and the mean degree to which nodes in a graph have a tendency to group together (average clustering coefficient) [85].

- **Synthetic graphs:** the probability distribution of the number of connections one node has to other nodes over the entire network (degree distribution), the sub-graphs that recur inside a single network or even across many networks (motif counts), and how closely nodes in a graph tend to group together (clustering coefficient) [104].

**Table 1:** A summary of representative deep graph generation models.

| Name | Model | Generation strategy | Sampling strategy | Graph type | | | | Application area | Year |
| --- | --- | --- | --- | --- | --- | --- | --- | --- | --- |
| | | | | Attributed | Weighted | Spatial | Temporal | | |
| GraphVAE [5] | VAE | Conditional | One-shot | ✓ | ✓ | | | Chemistry | 2018 |
| CGVAE [44] | VAE | Traverse-based | Sequential | ✓ | ✓ | | | Chemistry | 2018 |
| JT-VAE [61] | VAE | Traverse-based | Sequential | ✓ | ✓ | | | Chemistry | 2018 |
| Bresson and Laurent [43] | VAE | Traverse-based | One-shot | ✓ | ✓ | | | Chemistry | 2019 |
| NEVAE [65] | VAE | Traverse-based | Sequential | ✓ | ✓ | ✓ | | Chemistry | 2020 |
| HierVAE [114] | VAE | Conditional | One-shot | ✓ | ✓ | | | Chemistry | 2020 |
| NED-VAE [41] | VAE | Disentangled | One-shot | ✓ | ✓ | | | General | 2020 |
| Lim et al. [66] | VAE | Conditional | Sequential | ✓ | ✓ | | | Chemistry | 2020 |
| SGD-VAE [115] | VAE | Disentangled | One-shot | ✓ | ✓ | | | General | 2021 |
| DECO-VAE [57] | VAE | Disentangled | One-shot | ✓ | ✓ | | | Biology | 2021 |
| D2G2 [110] | VAE | Random | One-shot | ✓ | ✓ | | ✓ | General | 2021 |
| STGD-VAE [116] | VAE | Disentangled | One-shot | ✓ | ✓ | ✓ | ✓ | General | 2022 |
| MDVAE [54] | VAE | Conditional | Sequential | ✓ | ✓ | | | Chemistry | 2022 |
| D-MolVAE [109] | VAE | Disentangled | Sequential | ✓ | ✓ | | | Chemistry | 2022 |
| PGD-VAE [117] | VAE | Disentangled | One-shot | ✓ | ✓ | ✓ | | General | 2022 |
| GraphNVP [68] | NF | Traverse-based | One-shot | ✓ | ✓ | | | Chemistry | 2019 |
| GRF [88] | NF | Random | One-shot | ✓ | ✓ | | | Chemistry | 2019 |
| MoFlow [6] | NF | Traverse-based | One-shot | ✓ | ✓ | | | Chemistry | 2020 |
| GraphAF [81] | AR+NF | RL | Sequential | ✓ | ✓ | | | Chemistry | 2019 |
| GraphDF [69] | NF | RL | Sequential | ✓ | ✓ | | | Chemistry | 2021 |
| ChemSpacE [118] | LVM | Traverse-based | One-shot | ✓ | ✓ | | | Chemistry | 2022 |
| GCPN [14] | AR | RL | Sequential | ✓ | ✓ | | | Chemistry | 2018 |
| MolGAN [7] | GAN | RL | One-shot | ✓ | ✓ | | | Chemistry | 2018 |
| Mol-CycleGAN [73] | GAN | Conditional | One-shot | ✓ | ✓ | | | Chemistry | 2020 |
| GraphEBM [48] | EBM | Conditional | One-shot | ✓ | ✓ | | | Chemistry | 2021 |
| EDP-GNN [47] | Diffusion | Random | One-shot | ✓ | ✓ | | | General | 2020 |
| MOOD [119] | Diffusion | Conditional | One-shot | ✓ | ✓ | | | Chemistry | 2022 |
| DiGress [79] | Diffusion | Conditional | One-shot | ✓ | ✓ | ✓ | | General | 2022 |
| GDSS [8] | Diffusion | Random | One-shot | ✓ | ✓ | | | General | 2022 |
| GraphRNN [4] | AR | Random | Sequential | | | | | General | 2018 |
| GRAN [12] | AR | Random | Sequential | | | | | General | 2019 |
| MolecularRNN [52] | AR | RL | Sequential | ✓ | ✓ | | | Chemistry | 2019 |
| GRAM [104] | AR | Random | Sequential | | | | | General | 2019 |
| LFM [86] | AR | Random | Sequential | ✓ | ✓ | | | Chemistry | 2020 |
| BiGG [103] | AR | Random | Sequential | | | | | General | 2020 |
| STGG [120] | AR | RL | Sequential | ✓ | ✓ | | | Chemistry | 2021 |
| DeepGDL [85] | AR | Random | Sequential | ✓ | ✓ | | | Engineering | 2019 |

**Table 2:** Commonly-used evaluation metrics and datasets of deep graph generation.

| Task | Domain | Evaluation metrics | Datasets |
|---|---|---|---|
| Molecule generation | Chemistry | ◇ LogP
◇ QED
◇ SA
◇ DRD2 | ◇ QM9 [121]
◇ ZINC [122]
◇ MOSES [123]
◇ ChEMBL [124]
◇ CEPDB [125]
◇ PCBA [126]
◇ PDB [108] |
| Protein design | Biology | ◇ Perplexity
◇ Short/long-range contact | ◇ CATH [127]
◇ BRENDA [128]
◇ ProFold [115]
◇ Protein [129] |
| Social network modeling | Social science | ◇ Claw count
◇ Wedge count | ◇ DBLP-A [130]
◇ Travian [131]
◇ Ego [132]
◇ TwitterNet [133]
◇ CollabNet [130] |
| Physical simulation | Physics | ◇ Prediction error | ◇ N-body-charged, N-body-spring [111] |
| Transportation network design | Engineering | ◇ Betweenness centrality
◇ Broadcast centrality
◇ Node temporal correlation
◇ Global temporal correlation | ◇ METR-LA [134]
◇ PeMS-BAY [135] |
| Skeleton graph generation | Artificial intelligence | ◇ Mean time FID/IS
◇ Short FID/IS | ◇ Skeleton (Kinectics) [136]
◇ Skeleton (NTU-RGB+D) [137] |
| Power grid synthesis | Physics | ◇ Path length
◇ Network diameter
◇ Density | ◇ CUSPG [138] |
| Synthetic graph synthesis | N/A | ◇ Degree distribution
◇ Motif counts | ◇ Grid, Community [4]
◇ B-A, E-R, Scale-Free, Waxman [106] |

