# OpenReview forum: "A Survey on Deep Graph Generation: Methods and Applications"
_logconference.io/LOG/2022/Conference — LoG 2022 Poster_

### Official Review · Reviewer_Yf8R · 2022-09-22

**Overall Score:** 6
**Confidence:** 5

**Review:**

The paper presents a survey of deep generative models for graphs generation. In summary, it:

- formally describes the problem of graph generation;
- introduces a taxonomy to categorize different generative algorithms (latent variable (LV)-based and reinforcement learning (RL)-based);
- discusses the differences among similar learning problems on graphs;
- describes a set of applications where deep graph generation is currently applied;
- identifies core challenges to solve in order to take these methodologies to the next level.

I find this survey extremely pleasing to read; it is very well structured, written excellently, clear, and concise, with most of the relevant literature covered. In my opinion, this is exactly how a survey must be written, and I consider this a major strength of this work. Nonetheless, I find two key weaknesses that I'd like the authors to address in order to improve its contents, which I list below.

1) There are at least two other survey papers on graph generation ([1-2]) that are not cited in this work. I am not against another, but the authors should acknowledge their existence and discuss what this survey has to offer with respect to them (e.g. a different perspective?).
2) The survey seems to give little to no room to autoregressive (AR) approaches to graph generation, which are here presented just as a policy network for RL methods (section 3.2). GraphRNN, which is arguably among the works that started the field of deep graph generation, is only mentioned in the introduction. The AR approach doesn't fit in the proposed taxonomy; in fact, AR methods do not need to sample a latent code that summarizes the graph but can do so by sampling sequences of nodes (GraphRNN), edges [3], or a combination of both [4-5]. Nor do they need RL, unless the task is to optimize the graph structure (e.g. optimizing a molecular graph for drug-likeness), which is arguably outside of the scope of graph generation by itself. In summary, I think the AR approach fits the proposed taxonomy as a separate entity and should therefore be included. Otherwise, I'm eager to read the author's opinion as to why it doesn't fit.

Other minor points, in no particular order:
- In the introduction, the sentence "However, this assumption oversimplifies the underlying distributions of graphs and is thus too strong to satisfy" is a bit misleading. If it relaxes the problem, the assumption should be easier to satisfy, but not too strong to represent real-world graph distributions. Am I interpreting this sentence correctly?
- Arguably [6] is a citation that is missing, since it is probably the work where one-shot graph generation was introduced.
- Works like [7-8] should be mentioned in the context of scalability (Section 5) since they were among the first that scaled graph generation to graphs with thousands of nodes.
- In Section 5, I think that the generation of temporal graphs is one of the hardest challenges that researchers have to face. It is probably worth mentioning this research direction here (perhaps in the "Diverse graph types" subsection?).
- Line 131: "learns to" is italicized, but I think only "learns" should be.
- Perhaps the title "A Survey on Graph Generation: Methods and Application" is a bit misleading because it may appear to the reader as a survey on general graph generation methods, while only deep learning-based methods are considered.

In summary, I recommend the rejection of this paper in its current form. In my opinion, while very well structured and written, this work is missing: 1) an acknowledgment of previous similar work, and 2) the treatment of a whole streamline of graph generation methods; to be considered comprehensive. Nonetheless, I would be very happy to revise my decision in face of a convincing rebuttal.

I also understand there are page limits to be considered for this conference. If this round of reviews doesn't result in acceptance, I highly recommend the authors to try publishing in a journal, where there should be enough space to make this review exhaustive.

**EDIT**: the effort put in by the authors in the rebuttal convinced me to raise my score to a 6 (see my response to the authors below).

[1] Faez et al., "Deep Graph Generators: A Survey," IEEE Access 9 (2021)

[2] Guo et al., "A Systematic Survey on Deep Generative Models for Graph Generation", https://arxiv.org/abs/2007.06686 (2020)

[3] Bacciu et al., "Edge-based sequential graph generation with recurrent neural networks", Neurocomputing 416 (2020)

[4] Goyal et al., "GraphGen: A Scalable Approach to Domain-agnostic Labeled Graph Generation", WWW Proceedings (2020)

[5] Podda et al., "GraphGen-Redux: a Fast and Lightweight Recurrent Model for labeled Graph Generation", IJCNN Proceedings (2021)

[6] Simonovsky et al., "GraphVAE: Towards Generation of Small Graphs Using Variational Autoencoders", ICANN Proceedings (2018)

[7] Dai et al., "Scalable deep generative modeling for sparse graphs", ICML Proceedings (2020)

[8] Kawai et al., "Scalable Generative Models for Graphs with Graph Attention Mechanism", https://arxiv.org/abs/1906.01861

---

### Official Review · Reviewer_7TwZ · 2022-10-17

**Overall Score:** 6
**Confidence:** 4

**Review:**

Summary:
This paper provides a taxonomy survey on the latest methods in the research area of graph generation. Specifically, this survey paper gives a comprehensive review about latent variable based approaches including methods based on variational autoencoders, normalizing flows, generative adversarial networks, diffusion models, and reinforcement learning approaches. Besides, the paper introduces the applications and current challenges of existing graph generation methods.

Strengths:
(+) Generally, this paper gives a comprehensive and high-quality survey, including the most up-to-date graph generation methods. It provides a very good literature review and analysis about existing graph generation methods, which will be very useful for researchers in the research area.
(+) The writing of this paper is excellent and easy-to-follow.

Weaknesses:
(-) To my understanding, classifying existing graph approaches to "latent variable approaches" and "reinforcement learning approaches" is not very appropriate. Actually, reinforcement learning solely serves as a way to train generative models (of course can they be latent variable models), such as training sequential generation models to optimize molecular property, or passing gradients through discrete graphs with policy gradients in GCPN. Reinforcement learning itself does not provide any way to generate graphs. Hence, I personally suggest first classifying graph generative methods in the aspect of used generative models (autoregressive models, e.g., GraphRNN; VAE, e.g., JTVAE; flow, e.g., GraphAF; GAN, e.g., GCPN; diffusion, e.g., GDSS), then discussing different training methods for different tasks (vanilla data modeling, or target-oriented optimization).
(-) In Section 5, a recent explainable graph generation method [1] may be cited.

Overall, this paper provides an excellent literature survey about existing graph generation methods. However, the taxonomy of "latent variable approaches" and "reinforcement learning approaches" is not appropriate because reinforcement learning itself does not play a central role in deciding the way to generate graphs, which may cause confusion or misleading for readers. Hence, I vote for weak reject and will consider raising my score if authors can address this major concern.

Reference
[1] Tann, Wesley Joon-Wie, Ee-Chien Chang, and Bryan Hooi. "SHADOWCAST: Controllable Graph Generation with Explainability." (2020).

-----Post Rebuttal-----
I appreciate authors' hard work in their rebuttal. All my concerns have been well addressed. I have raised my score to weak accept.

---

### Official Review · Reviewer_umqA · 2022-10-22

**Overall Score:** 6
**Confidence:** 4

**Review:**

> (1) Summary

This paper provides a survey on graph generation, including a summary of methods and applications. This survey presents two mainstream methods called latent variable approaches and reinforcement learning approaches and focuses more on the first one. The authors summarize latent variable approaches into an encoder-sampler-decoder framework and introduce many related methods based on the framework. In addition to the methods and applications, the authors also highlight challenges and opportunities for future studies.

> (2) Strengths and weaknesses

Strengths:

- The writing is clear and easy to follow.
- The related problems part in Section 2 is interesting, linking graph generation to several related problems like link prediction and graph structure learning.
- Best on my knowledge, the summarized challenges and opportunities are clear and comprehensive.

Weaknesses / Questions:

- The summary of Deep Generative Models in Section 3.1.1 is not comprehensive. The authors include VAE, GAN, Flow, and Diffusion models here, but score-based [1] and energy-based [2] models are missed.
- I think the related problems part in Section 2 is very interesting. But there should be more references for readers if they are interested.
- Usually, we think graphs are permutation invariant. Can the mentioned methods preserve permutation invariance? Is this a challenge for graph generation?
- I think there are more related methods (and/or applications) the authors should consider.
For example, one application is 3D molecule generation (considering atom coordinates) [3][4].

> (3, 4) Recommendation

Currently, I recommend rejecting this paper, and my main concern is the lack of score-based and energy-based models. But I am happy to raise my score if the authors can address my concerns.

**-----Post Rebuttal-----**

I appreciate the authors' hard work in the rebuttal. My concerns have been addressed. I have raised my score to weak accept.

> (5, 6) Questions and feedback

See weaknesses.

> (7) 9 page

[1] Score-based generative modeling through stochastic differential equations

[2] GraphEBM: Molecular Graph Generation with Energy-Based Models

[3] Equivariant Diffusion for Molecule Generation in 3D

[4] An autoregressive flow model for 3d molecular geometry generation from scratch

---

### Meta-Review · Area_Chair_Rfpd · 2022-11-16

**Confidence:** 4
**Recommendation:** Accept

**Meta Review:**

This paper presents a high-quality survey on deep learning for graph generation, both in terms of methods and applications. The reviewers initially had some concerns about the comprehensiveness of the survey, but these were largely addressed in the rebuttal. The strong points, namely the high quality of the overall survey in terms of writing and presentation, and the excellent comparison of important representatives of related approaches (both in terms of similarities and differences), outweigh the negative points raised during the review process, and I think this survey will be a valuable contribution to the conference.

During the discussion period, the reviewers further encouraged adding the following points (possibly to the appendix) of the camera ready version: The authors are encouraged to expand on model details, discussions, analyses and summaries in the appendix -- for example, it was suggested that the authors should consider adding summary tables explaining/comparing existing methods, different applications or tasks, and different evaluation metrics.

---

### Decision · Program_Chairs · 2022-11-22

Accept (Poster)